# Non-Contact Current Sensing System Based on the Giant Magnetoimpedance Effect of CoFeNiSiB Amorphous Ribbon Meanders

**DOI:** 10.3390/mi15010161

**Published:** 2024-01-21

**Authors:** Zhen Yang, Zhenbao Wang, Mengyu Liu, Xuecheng Sun

**Affiliations:** 1School of Electronic and Information Engineering/School of Integrated Circuits, Guangxi Normal University, Guilin 541004, China; 1354724210@stu.gxnu.edu.cn (Z.W.); 1349375154@stu.gxnu.edu.cn (M.L.); 2Key Laboratory of Integrated Circuits and Microsystems, Education Department of Guangxi Zhuang Autonomous Region, Guangxi Normal University, Guilin 541004, China; 3Guangxi Key Laboratory of Brain-Inspired Computing and Intelligent Chips, School of Electronic and Information Engineering, Guangxi Normal University, Guilin 541004, China; 4Microelectronic Research & Development Center, School of Mechatronics Engineering and Automation, Shanghai University, Shanghai 200444, China

**Keywords:** current sensing system, GMI effect, probe, amorphous ribbon meanders

## Abstract

A sensitive non-contact sensing system based on the CoFeNiSiB amorphous ribbon giant magnetoimpedance (GMI) effect is proposed for current testing. The sensing system consists of a GMI probe, a sinusoidal current generator, a voltage follower, a preamplifier, a low-pass filter, and a peak detector. Four different GMI probes derived from amorphous ribbon meanders are designed and fabricated through MEMS processes. GMI probes were driven by a 10 MHz, 5 mA AC current. A permanent magnet was used to provide a bias magnetic field for the probe. The effect of the bias magnetic field on the output DC voltage was investigated. This non-contact current sensing system exhibits good sensitivity and linearity at a bias magnetic field *H*_bias_ = 15 Oe. The sensitivity can reach up to 24.2 mV/A in the ±1.5 A range.

## 1. Introduction

Magnetic sensors are a category of sensors that can convert magnetic field signals from physical phenomena into electrical signals for measurement. With the continuous progress of the detection of magnetic field signals, the demand for high-performance magnetic sensors has become more and more intense. This motivates more profound research on magnetic sensors. At the same time, the application of magnetic sensors is also becoming more and more extensive [1,2,3]. One of the relatively interesting applications is the current sensing field.

Modern power electronic systems often require current sensors to detect and regulate the current in the system. Current sensors can be divided into two types: contact measurement and non-contact measurement. Contact measurement is based on Ohm’s law that current is proportional to voltage. Usually, a small sampling resistor is connected in series to the circuit to convert the current into a voltage signal for measurement. The sample resistor itself affects the accuracy of the measured current, and contact measurements must be made with the circuit disconnected, which makes measurements cumbersome. The non-contact measurement based on Ampere’s law, where the magnetic field is proportional to the current, is not only convenient but also has the advantages of high sensitivity and low power consumption, which are favored by researchers. Among them, Hall sensors [4,5,6,7], giant magnetoresistive (GMR) sensors [8,9,10], giant magnetic impedance (GMI) sensors [11,12,13], etc., have attracted a lot of attention. Hall sensors are susceptible to temperature, and GMR sensors are difficult to fabricate, which makes the use of GMI sensors for current measurement increasingly popular.

The GMI effect is a phenomenon in which the AC impedance of a soft magnetic material changes significantly with an applied DC magnetic field [14,15,16,17]. The GMI effect has attracted a great deal of attention since its discovery, and research on GMI sensors has become a hot spot. In particular, some progress has been achieved in research on GMI current sensing applications [12,18,19]. The materials of the GMI probe of the current sensors mainly include amorphous wires [19,20,21], thin films [22,23,24], and amorphous ribbons [25,26,27], which all have good soft magnetic properties. Compared to amorphous wires, both thin films and amorphous ribbons can be patterned, which meets the diverse design needs of GMI probes. Due to the different preparation principles, GMI probes manufactured using thin films are more critical in terms of equipment and process, as these factors can significantly affect the quality of the thin film, which is not an issue for amorphous ribbons. The GMI probes produced using amorphous ribbons are not only simple to manufacture but also convenient to produce probes with similar properties in large quantities. Comparison reveals that amorphous ribbons are more suitable materials for the fabrication of GMI probes.

At present, the sensitive probes of most GMI current sensors are mainly composed of ring or stripe soft magnetic amorphous materials [11,13,18,27], which have limitations in sensitivity, size, or miniaturization. Aktham Asfour et al. utilized a 30 μm diameter GMI Co-based amorphous wire and curled it into a 2 cm diameter ring as a GMI probe. A biased magnetic field of 650 A/m is applied to the GMI probe to obtain an asymmetric GMI effect to detect the toroidal magnetic field generated by the measured current. Due to the smaller diameter of the amorphous wire, the detection area of the GMI probe is relatively small, which has a significant impact on the measurement accuracy. Bing Han et al. designed a novel-array-structured double-probe GMI current sensor by welding the strip amorphous ribbon with copper. A permanent magnet is used to provide a bias field for the probe. The double-probe output shows the best sensitivity and linearity at a bias magnetic field of 7.40 Oe. However, the welding process of this array probe is complicated, and the performance cannot be the same. In particular, there are some other structures [12,28], but their sensitive probes are still essentially strip structures. The sensing area is still small, and the sensitivity is insufficient. The meanders have been shown to significantly improve the GMI ratio and sensitivity compared to strip type [29,30,31,32,33] and can effectively utilize the element space to have higher spatial resolution [34], which is a more superior structure. Tao Wang et al. utilized multilayer meander films (NiFe/Cu/NiFe/Cu/Cr) to carry wide-frequency alternating currents (20 Hz–100 MHz) for quantitative measurements of DC currents [35]. It was found that the impedance of the current sensor was reduced in the presence of direct currents and decreased with increasing direct currents at medium frequency. But, at present, there is little research on detecting the current by adopting the amorphous ribbon meanders as the GMI probe; particularly, fewer studies have been conducted on the diagonal measurement of DC current with meander GMI probes.

In this study, we design and fabricate a meander diagonal current sensor based on commercial Co-based amorphous ribbon. The meander GMI probes are prepared by MEMS processes, which are connected to a stabilized crystal oscillator and driven by a high-frequency current generated by the oscillator. A permanent magnet provides a bias field for sensitive probes. The impedance of the sensitive probe changes due to the magnetic field generated by the measured current. Impedance signals are converted to voltage signals by subsequent circuit processing to measure current. This sensor is simple in construction and low cost, and it exhibits excellent linearity, stability, and reliability.

## 2. Design and Construction of Current Sensing System

### 2.1. Fabrication and Characteristics of Meander GMI Probes

The GMI probes are composed of amorphous ribbon meanders. The meanders are made of commercial cobalt-based amorphous ribbon. And the cobalt-based commercial ribbons with a nominal composition of CoFeNiSiB were purchased from Hebei King Do Electronic Co., Ltd. (Botou City, China) Compared with the amorphous ribbons of Metglas in the United States, the amorphous ribbon utilized in this design is not only superior to Metglas in some key parameters (such as maximum permeability) but also inexpensive and easy to purchase. The physical and magnetic properties of the ribbons are presented in Table 1.

The GMI probes were fabricated via the MEMS process. Details of the fabrication of the GMI probes have been reported elsewhere [13,16,17]. It can be briefly summarized as follows: (a) The amorphous strip was thinned and polished to bond it with the glass substrate. (b) The AZ4620 photoresist was spun on the amorphous ribbon. The parameters of the homogenizer were 500 r/min 15 s at low speed and 4000 r/min 60 s at high speed. The thickness of the photoresist after homogenization was 5 μm, and the UV exposure was 32 s. (c) The exposed sample was put into the AZ400K developing solution for 80 s and washed with deionized water for 20 s after the development was completed. (d) The etching solution was prepared according to the ratio of HCl:HNO_3_:H_2_O_2_:H_2_O = 1:2:4:8, and the sample was immersed in the etching solution by gradient etching for 60 s. The higher GMI ratio of the meanders compared to the strip type is due to the fact that the meanders have a more pronounced skin effect when a constant applied magnetic field is applied to the soft magnetic material. When the driving current frequency in the soft magnetic material is *f*, the skinning depth can be expressed as:(1)$$\delta =\sqrt{1/\pi f\sigma \mu_{\mathrm{eff}}}$$
where *σ* is the conductivity of the magnetic material, *μ*_eff_ is the effective permeability, and *f* is the driving current in the soft magnetic material. The impedance of meanders will change significantly with the external magnetic field *H*_ex_. Then, the impedance of the meanders can be calculated by [36,37,38,39,40]:(2)$$Z=\frac{\rho l}{2w\delta }+j\omega \left[L_{s}\left(\omega ,\mu_{\mathrm{eff}}\right)+L_{m}\left(\omega ,\mu_{\mathrm{eff}}\right)\right]$$
where *ρ* is resistivity, *δ* is skin depth, *L*_s_ is the self-inductance, *L_m_* is the mutual inductance, and *l* and *w* are the length and width of the ribbon, respectively. *ω* is the current angular frequency. The current sensing system is mainly based on the variation in the impedance of the meander GMI probe, which depends on the effective magnetic permeability *μ*_eff_.

As the AC frequency increases to the intermediate frequency regime (between ~100 kHz and a few MHz), domain wall displacement is damped and rotational magnetization starts to predominate [41]. When a driving current with a frequency of ~MHz is applied to the GMI probe, the domain wall displacement of the amorphous ribbon is inhibited, and the rotational magnetization starts to dominate. In this work, we investigated the performance of four different types of GMI probes. The meander GMI probes with three or six turns are utilized (as each “n” shape in the meanders represented one turn). Each of the four GMI probes was defined as a sample SA–SD. The parameters of the GMI probes are shown in Table 2.

The design of the four different GMI probes are shown in Figure 1. We have simultaneously investigated the performance of these four GMI probes. The dependence of the GMI ratio of the probes on the magnetic field is shown in Figure 2. It can be seen from Figure 2 that with the increase in the magnetic field, the GMI ratios of the four probes show a tendency of increasing and then decreasing, which is a typical characteristic of the GMI effect. It can be briefly summarized as before the application of external magnetic field, the domain wall displacement is hindered due to the pinning effect, which prevents the magnetization process. When the external magnetic field is applied to the material, the domain walls are freed from the pinning restriction, which makes the magnetization process easier [42], and the GMI ratio rises rapidly. When the external magnetic field continues to increase, the antimagnetized nuclei become antimagnetized domains in the presence of the external magnetic field, which leads to antimagnetization effects through domain wall displacement, and the overall magnetization efficiency of the material decreases, with a consequent decrease in the GMI ratio. It is observed that the anisotropic field of the ribbon is around 10 Oe, when the GMI ratio is at its maximum value. Among the four samples, sample SB has the highest GMI ratio of 94.7% (91.1% for SC, 86.3% for SA, 76.5% for SD). It is worth noting that the GMI ratio curve shows a monotonically decreasing trend when the applied magnetic field is larger than 10 Oe. It is obvious that the GMI probe has a good linear variation interval around 15 Oe. Therefore, we keep the operating point around 15 Oe by increasing the bias field.

### 2.2. Construction of Current Sensing System

The current sensing system consists of a GMI probe (detection of the magnetic field generated by the measured current), a sinusoidal current generator (generating high-frequency sinusoidal signals to drive GMI probes), a voltage follower (increasing the load-carrying capacity of the preamplifier system), a preamplifier (amplifying the output voltage signal of the GMI probe), a low-pass filter (filtering noise in amplifier circuits), and a peak detector (export of stable DC voltage signals). The vibration and signal processing circuit is shown in Figure 3.

The giant magneto impedance effects require high-frequency drive currents. A crystal oscillator-based sine signal generator can fulfill the demand. Due to the small temperature and time drift of the crystal oscillator and the very-high-quality factor Q, the crystal oscillator not only outputs accurate high-frequency signals (~MHz) but also has a high stability and a longer working period [43].

The frequency of the crystal oscillator depends on the equivalent inductance *L*_q_ and capacitance *C*_q_ of the equivalent series branch inside the crystal, and the equivalent capacitance *C*_0_ of the equivalent parallel branch. The specific parameters can be found in the datasheet of the crystal. *C*_L_ is the load capacitance of the crystal when it is operating in a parallel circuit, *C*_L_ = *C*_1_ × *C*_2_/(*C*_1_ + *C*_2_). The oscillator frequency *f*_0_ can be expressed by the following equation:(3)$$f_{0}=1/2\pi \sqrt{L_{q}\frac{C_{q}(C_{0}+C_{L})}{C_{q}+C_{0}+C_{L}}}$$

According to the parameters of the crystal oscillator, the load capacitance is generally taken as 10~100 pF. The load capacitance *C*_L_ = 33 pF in this design, the actual output frequency of the crystal oscillator, is about 10.1 MHz. The voltage signal generated by the crystal oscillator is converted into a constant sine current (*I*_ac_) of 5 mA peak to peak by resistor *R*_3_ and applied to the GMI probe. The voltage signal on the GMI probe is weak. In order to accurately transfer the voltage signal from the GMI probe to the subsequent circuit, the circuit is designed with a voltage follower as well as a preamplifier. The signal as well as the noise are amplified after passing through the preamplifier. A low-pass filter consisting of the OPA603AP is designed to filter out the noise signal. The peak detector, consisting of the ADA4817, outputs a DC voltage signal that is positively correlated with the amplitude of the GMI probe voltage. The DC voltage signal output from the peak detector is measured by a digital oscilloscope.

A high-frequency signal generated by a sinusoidal signal generator is applied to the GMI probe, and the impedance of the GMI probe changes due to the external magnetic field. Variation in impedance results in a change in the output voltage of the GMI probe. The voltage amplitude of the GMI probe can be modulated by an applied magnetic field (generated by the measured current). A schematic block diagram of current measurement is shown in Figure 4a. The DC power is used to drive the measurement system, and the oscilloscope is utilized to measure the output voltage of the GMI probe. A permanent magnet provides the bias field for the GMI probe and is positioned directly underneath the probe. The GMI ratio of the probe varies linearly near the bias magnetic field, which is shown as the bias point in Figure 1. A suitable bias field benefits the current measurement. The coil is the carrier of the current to be measured, and a transverse magnetic field *H*_i_ is generated in the coil when the measured current is applied. Magnetic field strength *H*_i_ varies with the measured current, and the impedance of the GMI probe varies with *H*_i_. The output of the signal processing circuit will output a voltage that is positively correlated with the change in impedance. In addition, in order to minimize the influence of the external magnetic field on the measurement data, a cubic magnetic shield is used to isolate the external magnetic field, as shown in Figure 4b. Figure 5 is a diagram of the actual test system; for convenient observation, the magnetic shield is not shown. Figure 5a–d are the fabricated GMI probes, and Figure 5e is the signal detection and processing unit.

## 3. Results and Discussion

### 3.1. Theory and Simulation

A distinctive characteristic of the GMI probe is that it is sensitive to magnetic fields. The variation in the external magnetic field can cause a significant change in the impedance of the GMI probe. The GMI ratio varies linearly near the bias magnetic field, which is an ideal measurement interval, as shown in Figure 2.

A 50-turns copper coil is used in the simulation to provide the DC magnetic field, and the GMI probe with a 3-turns structure is used inside. The impedance variation in the GMI probe and the magnetic field distribution around the probe were investigated. Figure 6 shows a three-dimensional view of the magnetic flux density distribution on the surface of the GMI probe in the GMI sensing system. A bar magnet is located below the coil to provide a DC-biased magnetic field for the probe. The current applied to the coil is the constant current to be measured. The GMI probe is located in the center of the coil, and the magnetic field lines are dense near the probe center, while the magnetic field lines are sparse at both ends of the probe. Therefore, the main sensing domain is in the middle of the probe, where the magnetic induction is significantly higher than at the ends. Due to the dissipation of the magnetic field in space and on the GMI detector, the magnetic field on the detector surface is actually 15 oe, which is lower than the calculated value [44,45,46].

According to Ampere’s law and the right-hand rule, an energized coil produces a steady magnetic field in the axial direction concentrated mainly at the center of the coil. Taking the bias field provided by the permanent magnet and the magnetic field generated by the current to be measured as the background magnetic field, it is found that the magnetic field on the surface of the GMI probe is lower than the background magnetic field. The magnetic field at the surface of the probe is actually lower than ideal because of the magnetic field dissipation in space, as well as on the GMI probe [47,48].

Keeping the background magnetic field conditions constant, the effect of the driving current frequency on the impedance in the GMI probe is analyzed. The impedance of the GMI probe increases as the frequency increases. And it shows a tendency to increase first and then decrease with the change in driving frequency. The impedance peaks at around 10 MH for all four samples. In particular, the change rate of impedance also showed a trend of first increasing and then decreasing. This phenomenon can be explained by the fact that the skinning effect on the GMI probe will be more pronounced when the background magnetic field is kept constant and the frequency of the driving current is gradually increased. It comes from an enhancement in the domain wall movement and domain rotation due to the influence of the applied constant magnetic field. The permeability of the GMI probe is altered, which results in the skinning depth becoming smaller, and its AC impedance changes significantly. As the driving frequency continues to increase, the magnetization rotation gradually dominates, leading to a decrease in the AC impedance [14,49], which can be explained by the ferromagnetic resonance theory. The impedance curve of the GMI probe is shown in Figure 7. Based on the simulation results, the driving current frequency is kept at 10 MHz in the experiments in order to improve the output voltage.

### 3.2. Experimental Section

In this work, the output voltage of the GMI probe is measured using a diagonal measurement method. The output voltage at the peak detection terminal is the converted DC voltage, which is directly proportional to the impedance value of the GMI probe and varies with the impedance of the probe. The current in the coil is measured indirectly.

In order to test the voltage output characteristics of the sensor, different test conditions are set to test the performance of the sensor. Figure 8 shows the dependence of the output voltage on the measured current for four GMI probes with and without permanent magnets, respectively. A bias magnet is positioned directly below the GMI probe to provide a bias magnetic field. The bias magnetic field is set at 15 Oe. It can be seen from Figure 8 that with the presence of a biased magnetic field, the output voltage varies more linearly with the measured current. The output DC voltage has good linearity when the measured current varies in the range of −1.5~1.5 A. The output voltage of the probe SB has the best linearity and the largest output voltage in the range of 84.8 mV to 157.41 mV (98.8 mV to 141.52 mV for SA, 91.3 mV to 148.91 mV for SC, 105.8 mV to 134.2 mV for SD). Without the bias field, the linearity of the output voltage is relatively poor. The output voltage varies linearly only within a small range, and it is discontinuous when the measured current direction changes.

To investigate the influence of different bias fields on the output voltage, the position of the permanent magnets is varied to provide various bias fields. Figure 9 shows the dependence of the output voltage on the measured current for four GMI probes with a DC-biased magnetic field from 1 to 18 Oe. For comparison purposes, the remaining parameters are maintained. However, because of the different positions of permanent magnets, the range of linear variation in all curves is not the same. Due to the placement of the permanent magnets, the bias field provided is different. The range of linear variation in the curves for the four GMI probes at various bias magnetic fields is different. For *H*_bias_ = 5, 13, 15, 18 Oe, the range of linear variation in the curve is −1.5~1.5 A (for *H*_bias_ = 1 Oe is −0.4~1.5 A, *H*_bias_ = 9 Oe is −1.5~1 A), beyond which the curve varies nonlinearly. Among the four GMI probes, sample SB has the most excellent output voltage characteristics. The output voltage of sample SB is higher as compared to the other three samples. When *H*_bias_ = 15 Oe, the output voltage range corresponding to sample SB is the largest, ranging from 84.8 mV to 157.41 mV (118.71 mV to 129.35 mV for *H*_bias_ = 1 Oe; 112.65 mV to 132.65 mV for *H*_bias_ = 5 Oe; 126.86 mV to 153.33 mV for *H*_bias_ = 9 Oe; 118.7 mV to 153.29 mV for *H*_bias_ = 13 Oe; 117.05 mV to 131.91 mV for *H*_bias_ = 18 Oe). Sample SD has the lowest output voltage compared to the other three samples. When *H*_bias_ = 15 Oe, the output voltage ranges from 105.8 mV to 134.21 mV (116.73 mV to 124.35 mV for *H*_bias_ = 1 Oe; 112.7 mV to 127.25 mV for *H*_bias_ = 5 Oe; 118.66 mV to 131.01 mV for *H*_bias_ = 9 Oe; 121.3 mV to 133.68 mV for *H*_bias_ = 13 Oe; 110.05 mV to 124.91 mV for *H*_bias_ = 18 Oe). That is more consistent with the measurements in Figure 2. The GMI ratio of four GMI probe samples is different. The GMI ratios of the four GMI probe samples are different, which results in various impedance ranges. And, for the same GMI probe, there are different results for various bias magnetic fields. Such a result can be explained by the fact that differences in the placement of the permanent magnet leads to differences in the bias field applied to the GMI probe. Therefore, when the coil current varies, the axial magnetic field generated by the coil makes the impedance of the GMI probe vary in different ranges. Then, the range of linear variation in output voltage is different. In general, the linearity of the output voltage and the range of variation are optimal when the bias field is 15 Oe.

To find out the nonlinear error of the sensor output voltage, several measurements have been completed in the current range of ±1.5 A. The adjusted R-squared is utilized to indicate the nonlinearity error of the test system output voltage (the adjusted R-squared closer to 1 indicates a lower nonlinearity error). The adjusted R-squared curves for different bias magnetic fields are shown in Figure 10. Different bias magnetic fields have a large effect on the nonlinearity error. In particular, the nonlinear error varies greatly for different probe types at the same bias magnetic field. It is clearly indicated that the adjusted R-squared values of the four probe types can reach more than 0.99 when the bias magnetic field is 15 Oe, and their nonlinear errors are at a low level, as expected.

Sensitivity is an important parameter to evaluate the performance of a sensor. The sensitivity of the four GMI probes is shown in Figure 11. The sensitivity of the four GMI probes showed a trend of first increasing and then decreasing. The peaks of sensitivity were observed at *H*_bias_ = 15 Oe. Among the four GMI probes, the sample SB has the highest peak of sensitivity up to 24.2 mV/A (14.20 mV for sample SA, 19.2 mV for sample SC, and 9.47 mV for sample SD). The sensitivity decreases as the bias field rises (when *H*_bias_ > 15 Oe), and this phenomenon is due to the fact that the GMI ratio decreases as the magnetic field continues to increase. But when the bias magnetic field is 1 to 13 Oe, the sensitivity is also lower than 15 Oe. This may be attributed to the large distance between the permanent magnet and the GMI probe, which means that the magnetic field applied to the GMI probe is unevenly distributed. As the probe is further away from the permanent magnet, the magnetic field becomes weaker and less uniform. And this inhomogeneity is expressed, not only in intensity but also in direction. It leads to the GMI probe not being uniformly magnetized in the longitudinal direction.

The noise will have an impact on the sensitivity and accuracy of the acquired signal, and the overall noise profile of the sensor is an important parameter to measure the performance of a sensor [50]. The main sources of the noise in the test system are the power supply as well as the amplifiers, which were measured separately. The noise curve of the test system in this design is shown in Figure 12. It can be clearly seen that the noise generated by the power supply is significantly higher than the noise generated by the amplifier. The noise of the test system is generally within a relatively lower level in the high-frequency range. Reducing the power supply and amplifier noise can further improve the sensitivity and accuracy of the test system.

## 4. Conclusions

A sensitive non-contact current sensing system based on the GMI effect was proposed. Four varying GMI probes derived from soft magnetic amorphous ribbon meanders were adopted. The steady DC current of the coil was measured through diagonal measurement. The experimental results showed that different bias magnetic fields have different effects on the output voltage as well as on the sensitivity of the GMI sensing system. It was found that when the bias magnetic field operating on the GMI probe was 15 Oe, the probe SB had the largest output voltage range of 84.8 mV to 157.41 mV and sensitivity of 24.2 mV/A within the measuring range of −1.5 A~1.5 A. The difference from previous the GMI current sensing system is that the GMI probe in the present text meanders. A permanent magnet was used to provide a bias magnetic field for the GMI probe in this design, enabling the output voltage of the GMI probe to vary linearly. Another advantage of using a permanent magnet to provide the bias field is that there is no need for an additional bias coil to provide the bias field, reducing power consumption. This design provides a way to optimize the novel current sensor.

## Figures and Tables

**Figure 1 micromachines-15-00161-f001:**
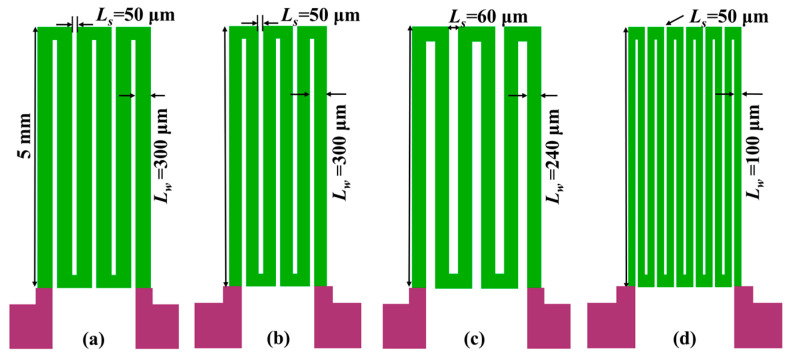
Schematic structure of four different GMI probes. (**a**) Schematic of the sample SA; (**b**) schematic of the sample SB; (**c**) schematic of the sample SC; (**d**) schematic of the sample SD.

**Figure 2 micromachines-15-00161-f002:**
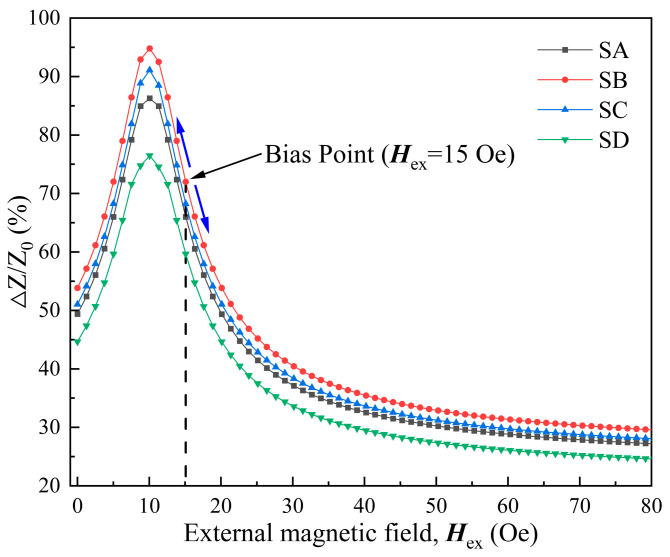
GMI characteristics of four different GMI probes.

**Figure 3 micromachines-15-00161-f003:**
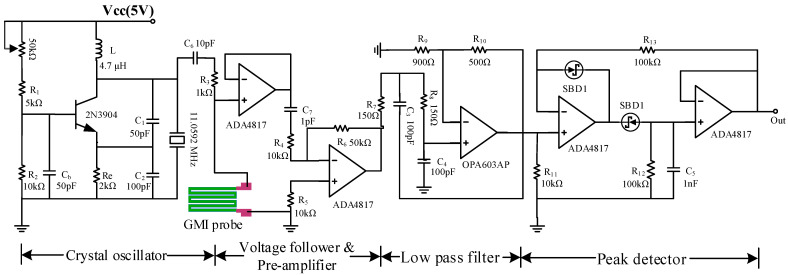
Vibration and signal processing circuit.

**Figure 4 micromachines-15-00161-f004:**
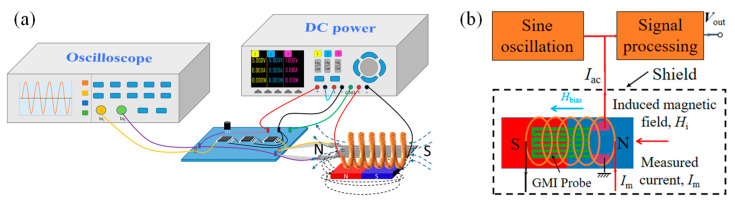
(**a**) Schematic diagram of diagonal structure current sensor and (**b**) the detail of GMI probe.

**Figure 5 micromachines-15-00161-f005:**
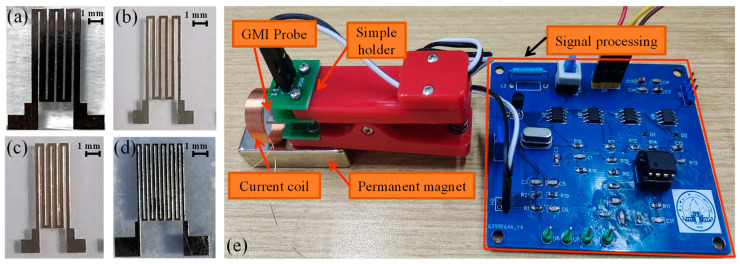
(**a**–**d**) are the fabricated GMI probes; (**e**) signal detection and processing unit.

**Figure 6 micromachines-15-00161-f006:**
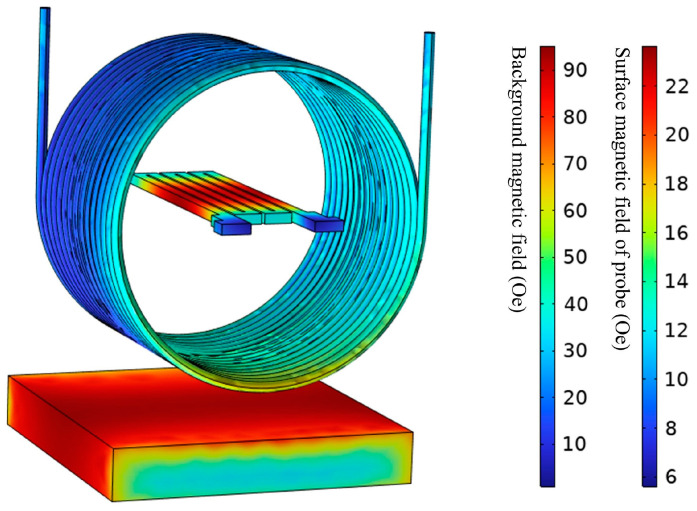
Magnetic field distribution around the GMI probe.

**Figure 7 micromachines-15-00161-f007:**
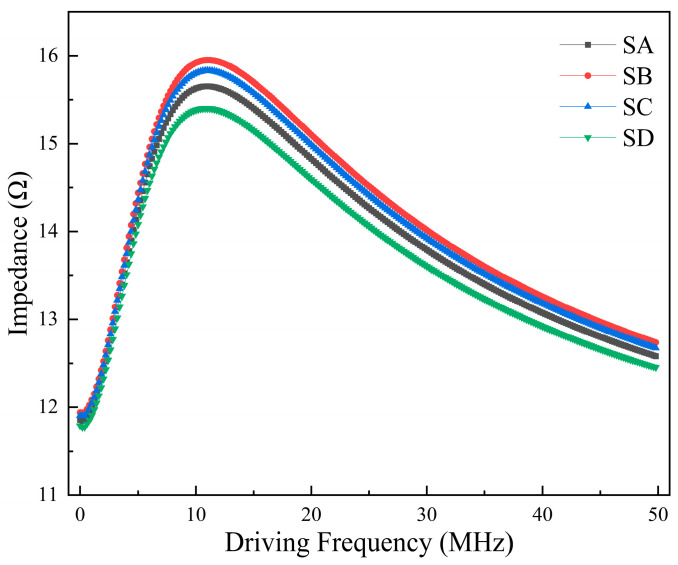
Dependence of impedance of four different GMI probes on driving frequency.

**Figure 8 micromachines-15-00161-f008:**
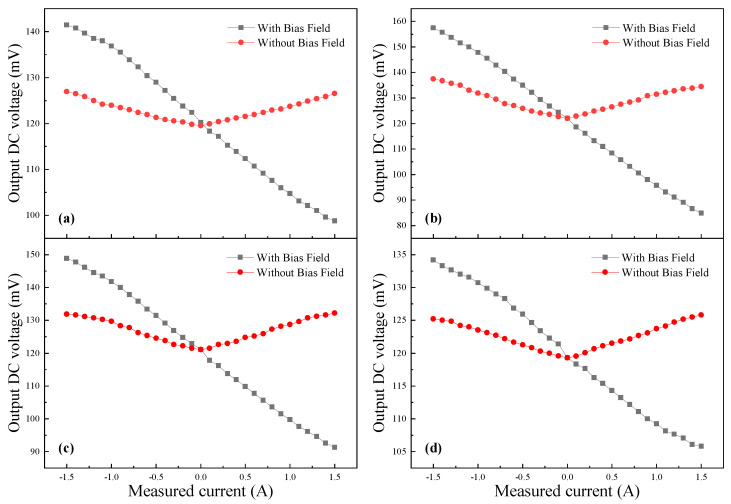
The dependence characteristics of the output voltage for four different GMI probes with and without bias field (**a**–**d**) are samples SA–SD.

**Figure 9 micromachines-15-00161-f009:**
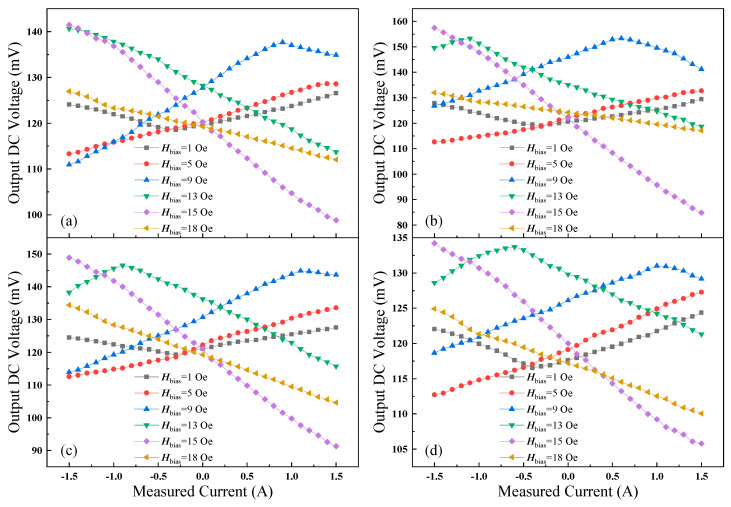
The dependence characteristics of the output voltage on the measured current for four different GMI probes. (**a**–**d**) are samples SA–SD.

**Figure 10 micromachines-15-00161-f010:**
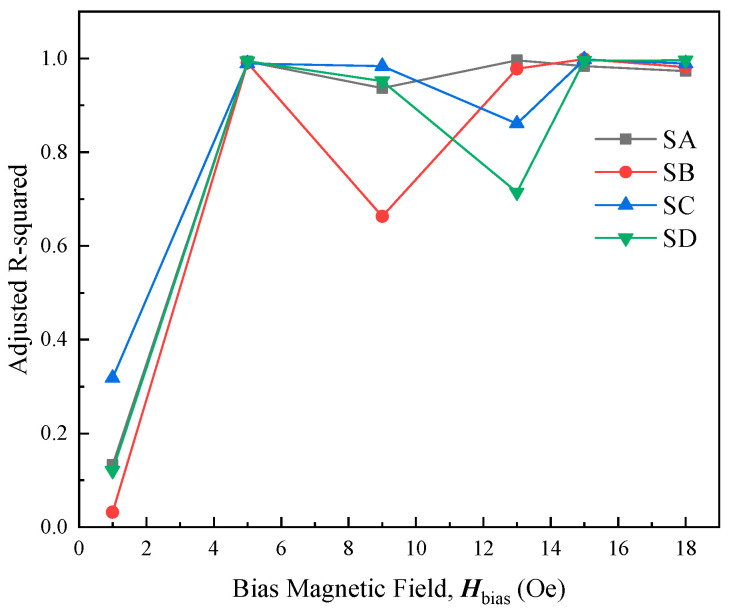
The dependence curves of different bias magnetic fields on the adjusted R-square.

**Figure 11 micromachines-15-00161-f011:**
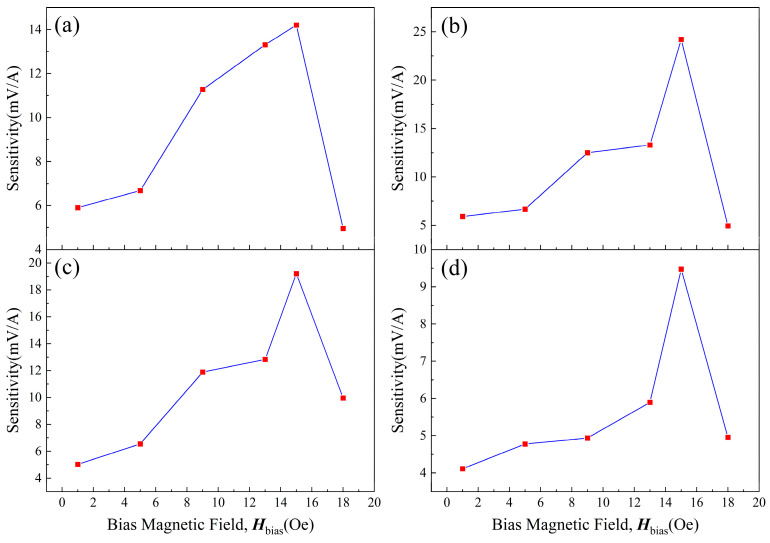
Sensitivity of four different GMI probes at various biased magnetic fields. (**a**–**d**) are samples SA–SD.

**Figure 12 micromachines-15-00161-f012:**
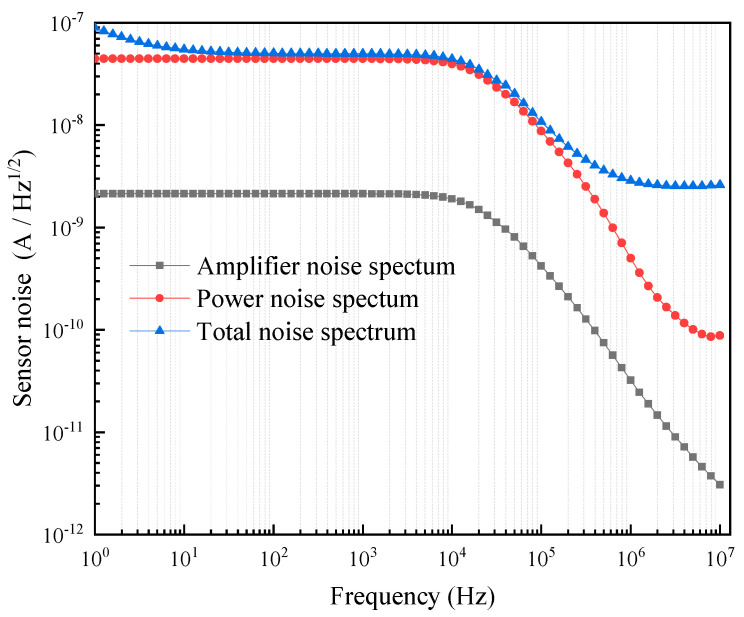
Noise spectrum distribution of the sensor.

**Table 1 micromachines-15-00161-t001:** Physical and magnetic properties of commercial ribbons.

Magnetic & Physical Properties		
	Hebei King Do	Metglas
Saturation Induction (T)	>0.55	0.5
Curie Temperature (°C)	205	200
Maximum Permeability (µ)	>1,200,000	1,000,000
Coercive Force (A/m)	<2.0	2
Density (g/cm^3^)	8.5	7.59
Crystallization Temperature (°C)	550	550

**Table 2 micromachines-15-00161-t002:** The parameters of the GMI probes.

	SA	SB	SC	SD
*L_s_* (μm)	50	50	60	50
*L_w_* (μm)	300	100	240	100
Turn	3	3	3	6

## Data Availability

Data are contained within the article.

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
