# Peer review of "Non-Contact Current Sensing System Based on the Giant Magnetoimpedance Effect of CoFeNiSiB Amorphous Ribbon Meanders"

_micromachines, 2024, doi:10.3390/mi15010161_

Round 1
Reviewer 1 Report
Comments and Suggestions for Authors
The authors report a sensitive non-contact current sensing system based on giant magnetoimpedance (GMI) effect. The test system mainly consists of a GMI sensitive element and a signal processing unit. And the processing unit consists of a sinusoidal current generator, a voltage follower, a preamplifier, a low-pass filter, and a peak detector. The GMI effect of the zigzag probe is investigated and the dependence of the output voltage of the test system with the measured current is analyzed.
Overall, this is an informative and interesting paper that provides a way of developing current sensors based on the GMI effect. The manuscript is well organized, and it can be fluently read by general audience. Some minor questions and suggestions should be properly addressed before the manuscript can be considered for its publication in journal:
(1)Please indicate in the title and abstract the material composition of the GMI sensitive element.
(2)The paper contains some grammatical problems and formatting errors, so please proofread it carefully before publication.
(3)In the experiment section,check the actual distribution of the magnetic field on the surface of the GMI probe after applying the bias field.
Comments on the Quality of English LanguageMinor editing of English language required.
Reviewer 2 Report
Comments and Suggestions for Authors
Comments to the Author
The non-contact current measurement on modern power electronic systems is not only convenient, but also has the advantages of high sensitivity and low power consumption, which is becoming increasingly application. This manuscript showed a sensitive non-contact current sensing system based on the GMI effect. The difference from previous GMI current sensing system is that the GMI probe in the present text is meanders, which is convenient in certain application. What’s more, a permanent magnet is used to provide a bias magnetic field for the GMI probe in this design, enabling the output voltage of the GMI probe to vary linearly. Another advantage of using permanent magnet to provide the bias field is that there is no need for an additional bias coil to provide the bias field, reducing power consumption. This design provides a way to optimize the novel current sensor. Thus, I suggest its publication. However, there are some questions in the article, which should be clarified before further consideration for publication.
1. Line 102, table 1, compared with previous commercial ribbons, what is the progressiveness of the amorphous ribbon meanders proposed in this paper? The author should create a table for comparison.
2. Line 104,this article should also briefly introduce the fabrication of the GMI probes, even if described elsewhere.
3. Try to explain the physical mechanisms that cause the four structural differences as shown in Figure 2, Figure 7, Figure 8, and Figure 9.
4. Please give the non-linear errors as shown in Figure 8, and Figure 9.
5. Noise is an important parameter to evaluate the performance of a sensor. Please provide the noise test results.
Round 2
Reviewer 2 Report
Comments and Suggestions for Authors
This paper is much improved since my first review. I don't have more suggestion.